

# Attribution of recent ozone changes in the Southern Hemisphere mid-latitudes using statistical analysis and chemistry-climate model simulations

Guang Zeng[1], Olaf Morgenstern[1], Hisako Shiona[2], Alan J. Thomas[3], Richard R. Querel[3], and Sylvia E. Nichol[1]

[1]National Institute of Water and Atmospheric Research, Wellington, New Zealand
[2]National Institute of Water and Atmospheric Research, Christchurch, New Zealand
[3]National Institute of Water and Atmospheric Research, Lauder, New Zealand

*Correspondence to:* Guang Zeng (guang.zeng@niwa.co.nz)

**Abstract.**

Ozone ($O_3$) trends and variability from a 28-year (1987-2014) ozonesonde record at Lauder, New Zealand, have been analysed and interpreted using a statistical model and a global chemistry-climate model (CCM). Lauder is a clean rural measurement site often representative of the Southern Hemisphere (SH) mid-latitude background atmosphere. $O_3$ trends over this period at this location are characterised by a significant positive trend below 6 km, a significant negative trend in the tropopause region and the lower stratosphere between 9 to 15 km, and no significant trend in the free troposphere (6-9 km) and the stratosphere above 15 km. We find that significant positive trends in lower tropospheric ozone are correlated with increasing temperature and decreasing relative humidity at the surface over this period, whereas significant negative trends in the upper troposphere and the lower stratosphere appear to be strongly linked to an upward trend of the tropopause height, associated with increasing greenhouse gases. Relative humidity and the tropopause height also dominate $O_3$ variability at Lauder in the lower troposphere and the tropopause region, respectively. We perform an attribution of these trends to anthropogenic forcings including $O_3$ precursors, greenhouse gases (GHGs), and $O_3$ depleting substances (ODSs), using CCM simulations. Results indicate that changes in anthropogenic $O_3$ precursors contribute significantly to stratospheric $O_3$ reduction, changes in ODSs contribute significantly to tropospheric $O_3$ reduction, and increased GHGs contribute significantly to stratospheric $O_3$ increases at Lauder. Methane ($CH_4$) likely contributes positively to $O_3$ trends in both the troposphere and the stratosphere, but the contribution is not significant at the 95% confidence level over this period. An extended analysis of CCM results covering 1960-2010 (i.e. starting well before the observations) reveals significant contributions from all forcings to $O_3$ trends at Lauder, i.e., increases of GHGs and the increase of $CH_4$ alone all contribute significantly to $O_3$ increases, net increases of ODSs lead to $O_3$ reduction, and increases of non-methane $O_3$ precursors cause $O_3$ increases in the troposphere and reductions in the stratosphere. This study suggests that a long-term ozonesonde record obtained at a SH mid-latitude background site (corroborated by a surface $O_3$ record at a nearby SH mid-latitude site, Baring Head, which also shows a significant positive trend) is a useful indicator for detecting atmospheric composition and climate change associated with human activities.



# 1 Introduction

Ozone ($O_3$) is an important trace gas in the Earth's atmosphere, playing a central role in atmospheric chemistry and the radiation budget. Stratospheric $O_3$ prevents harmful UV light from reaching the Earth's surface and is also a source of tropospheric ozone
via cross tropopause exchange. Tropospheric $O_3$ is formed by photochemical production in the presence of nitrogen oxides and volatile organic compounds (VOC). It controls the atmospheric oxidizing capacity through photolysis and subsequent reaction with water vapour to produce the hydroxyl radical (OH). Tropospheric $O_3$ is inhomogeneously distributed and highly dependent on the distributions of its precursors and on tranport. High levels of surface $O_3$ have adverse effects on human health and vegetation. Since the industrial era, surface $O_3$ in the Northern Hemisphere (NH) has increased substantially (e.g., Volz
and Kley, 1988; Thompson, 1992) due to human activities. By contrast, surface $O_3$ in the Southern Hemisphere (SH) is more stable due to limited land and a general lack of precursor emissions. Here, surface $O_3$ is the result of stratosphere-troposphere exchange (STE) and methane ($CH_4$) oxidation.

In the SH, stratospheric $O_3$ changes, which are dominated by Antarctic ozone depletion, show significant negative trends especially at mid- to high-latitudes from the 1980s to the end of the 1990s (WMO, 2011). Reductions in stratospheric halogen
due to the implementation of the Montreal Protocol would increase both tropospheric and stratospheric $O_3$; indeed, observations show that stratospheric $O_3$ between 35-45 km has shown no trend since the late 1990s (Figure 2-5 in WMO (2014)). However, ozone recovery is modulated by climate change. Model simulations show that recovery will occur between 2030 and 2070 (depending on region), but there may be no sustained recovery in the tropics where climate change accelerates the export of $O_3$ to higher latitudes, reducing $O_3$ (Eyring et al., 2010). As a consequence, future changes of stratospheric $O_3$ will
significantly impact tropospheric $O_3$ and potentially air quality (e.g., Zeng et al., 2010).

Understanding past $O_3$ changes and projecting the future $O_3$ evolution are of great importance to assessing future environmental changes. To cover its spatial and temporal variability, an extensive global ozone monitoring network exists measuring total column ozone, vertical profiles, and surface ozone. These measurements cover years to decades. Oltmans et al. (2013) analyse tropospheric $O_3$ trends from a large suite of surface $O_3$ and ozonesonde measurements for the past 20-40 years and find
that there is no significant overall change in tropospheric $O_3$ in the NH and tropics; they attribute this no-growth to controls on $O_3$ precursor emissions after earlier increases of tropospheric $O_3$. However, in the SH subtropics and mid-latitudes, near-surface $O_3$ exhibits a significant positive trend since the 1980s but there have been no trends in the free and upper troposphere; reasons for this behaviour are not understood.

In this paper, we present the updated ozonesonde record covering 1987-2014 from Lauder, New Zealand (45ºS, 170ºE, 370 m
above sea level), a clean rural site that is representative of the SH mid-latitude background atmosphere. Parts of the time series have been shown previously by Oltmans et al. (2006, 2013). Here, we focus on analysing and interpreting the tropospheric and lower stratospheric $O_3$ changes using a simple statistical model and global chemistry-climate model (CCM) simulations, and identify the underlying dynamical and chemical changes that could contribute to the observed $O_3$ trends at this location, which



also represent the evolution of background $O_3$ in the SH mid-latitudes in general. Although we mainly analyse the ozonesonde record at Lauder, the surface $O_3$ record at another SH mid-latitude site, Baring Head, New Zealand, will also be presented for comparison. We describe the ozonesonde time series, the statistical model used, and the chemistry-climate model simulations in the following section. Results are presented in Sect. 3, and conclusions are drawn in Sect. 4.

## 2    Ozone records, statistical method, and chemistry-climate model simulations

Weekly Electrochemical Cell (ECC) ozonesondes, launched at Lauder since August 1986, measure profiles of $O_3$, temperature, pressure and winds from the surface to about 35 km. The ozonesondes used are ECC EN-SCI IZ-series operating with a 0.5% buffered potassium iodide (KI) cathode solution (Boyd et al., 1998). Corrections are applied to the ozonesonde values above 200 hPa to account for pump efficiency degradation at low pressures (Bodeker et al., 1998). The integrated $O_3$ profile is compared

to the total column of ozone measured by the Dobson spectrophotometer at Lauder; the relative difference is typically less than 5% (Bodeker et al., 1998). For this analysis, following Oltmans et al. (2013), we calculate vertically averaged $O_3$ mixing ratios for eight layers, namely, 0-1.5 km, 1.5-3 km, 3-6 km, 6-9 km, 9-12 km, 12-15 km, 15-20 km, and 20-25 km. Separate linear trends are then calculated for $O_3$ anomalies at each layer and over the whole period.

We construct a regression model to identify the dominant factors that are associated with $O_3$ variations and trends. The

15 regression model includes eight time-varying forcings, i.e., the Solar Index ($SI$) which captures solar variability and is defined by the solar radio flux at F10.7 cm (data source: https://www.esrl.noaa.gov/psd/data/correlation/solar.data), the Multivariate El Niño Southern Oscillation Index ($MEI$) (https://www.esrl.noaa.gov/psd/enso/mei/), the Quasi-Biennial Oscillation at 30 hPa and 10 hPa, respectively ($QBO_{30}$ and $QBO_{10}$) (http://www.esrl.noaa.gov/psd/data/climateindices/list/), relative humidity at the surface ($RH$), stratospheric temperature averaged over 20-25 km ($T_{strat}$), tropopause height ($HT$), and the effective

equivalent chlorine loading ($Cl_y$). RH and $T_{strat}$ are measured together with $O_3$ by the ozonesondes. Tropopause height is calculated using the 150 ppbv $O_3$ chemical tropopause, defined using observed $O_3$ at Lauder. $Cl_y$ is the total chlorine loading at 20 km over Lauder, which is taken from a transient CCM simulation assuming the WMO A1 scenario (WMO, 2011). The regressed ozone anomaly is expressed as

$$Ozone(t) = A*t + a_1*SI(t) + a_2*MEI(t) + a_3*QBO_{10}(t) + a_4*QBO_{30}(t)$$
$$+ a_5*RH(t) + a_6*T_{strat}(t) + a_7*HT(t) + a_8*Cl_y(t), \qquad (1)$$

of which $A*t$ is the linear component, and $a_{1-8}$ are the regression coefficients for corresponding functions. All forcings are standardised but not de-trended. We note that interdependencies and correlations, e.g., between stratospheric temperature, relative humidity at surface, and tropopause height as regression functions, cannot be excluded. Observations as well as basis functions are monthly mean data.

We perform and analyse a suite of sensitivity simulations with different model configurations aimed at investigating the

30 factors that potentially contribute to observed $O_3$ trends at Lauder, e.g., $CH_4$ and other $O_3$ precursors, greenhouse gases (GHGs), and ozone depleting substances (ODSs) over the period of the time series (1987-2014). The model simulations we





analyse here are from the National Institute of Water and Atmospheric Research-United Kingdom Chemistry and Aerosols (NIWA-UKCA) model, which comprises a detailed representation of stratospheric and tropospheric chemistry and is optionally run in atmosphere-only mode or coupled to a deep-ocean model. Detailed descriptions of the model are given by Morgenstern et al. (2009, 2017), O'Connor et al. (2014), and Zeng et al. (2015). We only give a brief description of the version used here. The

background climate model is an early version of HadGEM3-A (Hewitt et al., 2011) at a horizontal resolution of $3.75° \times 2.5°$, with 60 levels from the surface to 84 km. Chemistry combines the stratospheric mechanism described by Morgenstern et al. (2009) and the tropospheric mechanism used by Zeng et al. (2008). Eight primary pollutants are emitted at the surface, namely nitrogen oxides ($NO_x$), carbon monoxide (CO), formaldehyde ($CH_2O$), ethane ($C_2H_6$), acetaldehyde ($CH_3CHO$), propane ($C_3H_8$), acetone ($CH_3COCH_3$), and isoprene ($C_5H_8$). We assume uniform, constant lower boundary conditions for $N_2O$, $CH_4$,

$H_2$, and organic halogen compounds. Chlorine and bromine source gases are lumped, and we only explicitly model $CFCl_3$, $CF_2Cl_2$, $CH_3Br$, $CH_2Br_2$, and $CHBr_3$ here. Aerosol and aerosol precursor emissions are included. Lightning emissions of $NO_x$ are parameterized as a function of cloud top height that is linked to convection in the model (Price and Rind, 1992, 1994). The model uses the FAST-JX interactive photolysis scheme (Neu et al., 2007; Telford et al., 2013) with a correction added above 60 km for photolysis occurring at wavelengths shorter than 177 nm (Lary and Pyle, 1991).

All simulations used in this study are listed in Table 1; they are part of the Chemistry Climate Model Initiative (CCMI) model simulations (Eyring et al., 2013; Morgenstern et al., 2017). The model data can be downloaded from the British Atmospheric Data Centre (BADC) (http://blogs.reading.ac.uk/ccmi/badc-data-access). We keep the experiment identifications as defined by Eyring et al. (2013) and Morgenstern et al. (2017) for CCMI. REF-C1 is a hindcast experiment using prescribed observed monthly mean sea surface temperatures (SSTs) and sea ice (HadISST Rayner et al., 2003)) over the period of 1960-2010. Other

forcings, e.g., GHGs, ODSs, and aerosol and aerosol precursor emissions follow state-of-knowledge historical evolutions from 1960 to 2010 (Morgenstern et al., 2017). The anthropogenic emissions of $O_3$ precursors are inter-annually varying following the MACCity scenario from 1960 to 2010 (Granier et al., 2011), and biogenic emissions are prescribed following the MEGAN2.1 (Guenther et al., 2012) dataset with no inter-annual variation. $CH_4$ mixing ratios are prescribed at the surface, and the same $CH_4$ scenario is used in both chemistry and radiation. In this set up, chemistry is interactive, in that calculated $O_3$ is fed into

the radiation scheme. We consider REF-C1 to be the state-of-knowledge experiment of the evolution of chemical composition from 1960 to 2010. We also performed a simulation with non-methane $O_3$ precursor and emissions fixed at the 1960 level (SEN-C1-fEMIS) to assess the impact of increases in these emissions since 1960. Here we analyse the results between 1987 and 2010, to coincide with the Lauder ozonesonde measurements. In addition, we specify a stratospheric $O_3$ tracer ($O_3S$) to examine the impact of stratosphere-troposphere exchange (STE) on tropospheric $O_3$ in experiment REF-C1. $O_3S$ is defined as

having the same chemical destruction as the "normal" chemical $O_3$ tracer but no chemical production in the troposphere, to account for the $O_3$ that is originated from the stratosphere.

The second set of simulations, listed in Table 1 (REF-C2, SEN-C2-fODS, SEN-C2-fGHG, and SEN-C2-fCH4) are used to attribute $O_3$ changes to ODSs and GHGs, respectively. These simulations are carried out using the coupled atmosphere-ocean configuration. The simulations cover the period of 1960-2100, and follow the WMO (2011) A1 scenario for ODSs and the

Representative Concentration Pathway (RCP) 6.0 (Meinshausen et al., 2011) for other GHGs, tropospheric $O_3$ precursors, and





aerosol and aerosol precursor emissions. For anthropogenic $O_3$ precursor emissions, we use MACCity emissions until 2000, followed by RCP 6.0 emission, as recommended for CCMI simulations. Unfortunately, there is a discontinuity in $NO_x$ emissions when this transition occurs (Morgenstern et al., 2017), which will impact the $O_3$ trend calculation. The three sensitivity simulations differ from the REF-C2 simulations in that ODSs, GHGs, and $CH_4$, respectively, are fixed at their 1960 levels.

The discontinuity in $NO_x$ emissions exists in all three sensitivity simulations, therefore, we anticipate that its effect will be similar across all simulations. We analyse results from 1987 to 2014, which cover the Lauder ozonesonde time series. The time evolutions of $CH_4$, $CO_2$, $N_2O$, $Cl_y$, and $Br_y$, and the surface emissions of $NO_x$ and CO are displayed in Fig. 1.

## 3 Observed ozone variabilities and trends at Southern Hemisphere mid-latitudes

### 3.1 Ozone variability

Fig. 2 shows the deseasonalised $O_3$ anomalies at the eight layers from the surface to the lower stratosphere, and the respective regressed $O_3$ anomalies. Large interannual variations appear in the observed $O_3$ anomalies. The $O_3$ regression captures much of the variability in the stratosphere, but is less accurate at capturing variations in tropospheric $O_3$ in general. The major contributing variables to the regressed $O_3$ anomalies are shown in Fig. 3, and are summarised in Table 2. At the surface, the $O_3$ anomaly is reasonably well captured by the regression which is dominated by relative humidity; this may indicate

that stratospheric intrusions play a significant role in controlling surface $O_3$ variations at Lauder, as RH is considered an effective indicator of such intrusions (e.g., Cristofanelli et al., 2006; Stohl et al., 2000). Around and immediately above the tropopause (i.e. 9-15km), the $O_3$ anomalies largely follow the tropopause height, which reflects that $O_3$ in this region is mainly controlled by changes in dynamics. Such dynamical changes also influence $O_3$ variability throughout the troposphere and lower stratosphere (see Table 2). In the free troposphere, between 1.5km and 9km, $O_3$ variations project onto a number of factors

including tropopause height, RH, temperature, and the MEI, with RH playing a dominant role in the lower troposphere. There is a notable influence of ENSO on ozone variations in the region between 3 and 9 km. In the lower stratosphere, $O_3$ is mainly controlled by the QBO (at both 30 hPa and 10 hPa), and by stratospheric temperature. Note that when considering regression factors, we do not remove dependencies between variables.

Time series of some of the meteorological and climate variables used in the regression are shown in Fig. 4. Except for the

25 MEI time series, all other variables are co-measurements with the $O_3$ measurements at Lauder. Surface temperature is also shown which is not part of the regression function. In order to directly assess the relationship between these variables and $O_3$ variability, we calculate correlation coefficients of these variables with $O_3$ anomalies for each layer (see Table 3). This analysis shows clearly that RH has a strong anti-correlation with the $O_3$ anomalies close to the surface, and the correlation reduces at higher altitudes, implying a role for deep stratospheric intrusion in near-surface $O_3$ variability as mentioned above.

Surface temperature anomalies and $O_3$ anomalies in the troposphere are positively correlated, but are anti-correlated in the stratosphere. Likewise, stratospheric temperature anomalies and stratospheric $O_3$ anomalies are correlated. There are rather weak correlations between ENSO (expressed as MEI) and $O_3$ anomalies, with the largest influence from ENSO in the free troposphere at this mid-latitude location. QBO at 30 hPa is mostly anti-correlated with ozone anomalies in the stratosphere.



## 3.2 Ozone trends

The linear trends in observed deseasonalised $O_3$ anomalies calculated for each layer are listed in Table 4, and are also shown in Fig. 2. Significant positive linear trends ( 0.04 ppbv/yr) are found from the surface up to 6 km over the 28 year period (1987 to 2014). No significant trend is calculated for 6-9km. Above 9 km, trends become significantly negative, but with an insignificant

negative trend obtained for 20-25 km. To examine if there are any synergies between the trends in $O_3$ and the dominant meteorological and climate variables, the linear trends of the meteorological and climate variables over this period are also calculated and are shown in Fig. 4; it clearly shows that there are significant linear trends in MEI, RH, temperatures, and tropopause height anomalies over the period of 1987 to 2014, with significant positive trends in surface temperature ($0.02\pm0.01$K/yr), and the tropopause height ($14.2\pm5.1$ m/yr), and significant negative trends in RH ($-0.25\pm0.06$%/yr), stratospheric temperatures

($-0.13\pm0.01$K/yr), and MEI ($-0.03\pm0.01$/yr) (Fig. 4). Significant trends in these variables may indicate some shifts in climate variability, and climate change, which are mainly induced by increasing GHGs (e.g., Mitchell et al., 1995; Santer et al., 1996). Indeed, the increase in surface temperature and the large decrease in stratospheric temperature are the result of a modified radiation balance. Temperature changes in turn result in dynamical changes, such as changes in tropopause height as shown in Fig. 4, likely due to stratospheric cooling. RH at the surface shows a significantly strong negative trend; this is typically linked

to increased deep stratospheric intrusion. MEI shows a moderate negative trend which can be explained by a shift in climate variability. We also calculate $O_3$ trends over the period of 1987 to 2010, and obtain slightly larger significant positive $O_3$ trends in the troposphere (Table 4) than those from the period of 1987 to 2014.

Our trend calculation is in broad agreement with Oltmans et al. (2013), who calculated a trend of 0.15 ppbv/yr in $O_3$ from the surface to the free troposphere over the period of 1986-2010, and no significant trend above 500 hPa, although they did

not include $O_3$ trends in the stratosphere in their study. Oltmans et al. (2013) did not elaborate on what exactly drives the significant positive trends in the lower troposphere at the SH mid-latitude site Lauder. There is no significant trend in the upper troposphere. They suggested that such a pattern in tropospheric $O_3$ trends does not reflect a possible increase in stratosphere-troposphere exchange (STE), as increasing STE would lead to an increase in $O_3$ in the upper and free troposphere. From our analysis based on regression/correlation, the deep stratospheric intrusions may play a role in the lower tropospheric $O_3$ trends,

reflected in the significant negative trend in RH over the period of 1987 to 2014 (Fig. 4). However, we cannot derive simple and direct links between changes in RH, deep stratospheric intrusions, and the observed trends in near-surface $O_3$. In the tropopause region, the change in tropopause height clearly drives the $O_3$ variation and trend, i.e., significant upward trends in tropopause height imply significant negative $O_3$ trends between 9 to 15 km (Figs. 2 and 4). In the lower stratosphere above 15 km, negative $O_3$ trends become weaker as the stratospheric temperature plays an increasingly dominant role in controlling changes in $O_3$,

i.e., stratospheric cooling (Fig. 4) is likely to slow down chemical $O_3$ destruction, leading to an increase in stratospheric $O_3$. This would cancel out the decrease in $O_3$ resulting from the tropopause height increase. Similarly, the insignificant trend in $O_3$ between 6 and 9 km may be the result of cancellations between the increase of $O_3$ in the lower troposphere and the decrease of $O_3$ in the tropopause region. In Sect. 4, we analyse a set of sensitivity simulations designed to assess the contributions from





some anthropogenic forcings (ODSs, GHGs, and $O_3$ precursors), that are known to have caused substantial changes in recent decades to $O_3$ trends.

### 3.3 Baring Head surface $O_3$

In addition to the ozonesonde record at Lauder, we examine the surface $O_3$ time series from Baring Head (41.4S, 174.9E, 85m) which is also a SH mid-latitude background station located in central New Zealand. In southerly wind situations Baring Head, a coastal site, is exposed to clean marine air. The surface $O_3$ measurements at Baring Head started in 1991; the record is largely continuous except for some missing data in 2005. The surface $O_3$ measurements have been made using ultraviolet (UV) photometric ozone analysers (Dasibi model 1003-PC from 1991-2004, and Thermo Electric Corporation TEI49i from 1995), with air drawn from 5 metres above the ground through a dedicated teflon tube with a 0.5 $\mu$ teflon filter on the inlet to exclude aerosols. Hourly data from 1992 to 2015 are used to calculated the surface $O_3$ anomaly and the trend. The data are also deseasonalised.

Shown in Fig. 5, a significant positive trend is calculated in surface $O_3$ (0.057 ppbv year$^{-1}$) at Baring Head, which is similar to the trend found at Lauder (0.04−0.06 ppbv year$^{-1}$), but the positive trend calculated from near-surface ozonesonde data at Lauder exhibits a much larger uncertainty range. The significant positive trend in surface $O_3$ at Baring Head, together with the significant positive trends in Cape Grim and Cape Point surface $O_3$ calculated by Oltmans et al. (2013), confirms that the positive surface $O_3$ trends in the SH mid-latitudes are robust. The insignificant positive trend in surface $O_3$ at Baring Head calculated by Oltmans et al. (2013) over the period of 1991-2010 is due to the shorter data record compared to the record used here.

## 4 Attribution of ozone trends using chemistry-climate model simulations

### 4.1 Modelled ozone trends at Lauder

In order to attribute the observed $O_3$ trends at Lauder in a more quantitative way, we compare the observations to model results. Although the model has a relatively coarse resolution, the measurement site is representative of SH mid-latitudes which are characterized by relatively large-scale variability in $O_3$. Firstly, we show in Fig. 6 $O_3$ volume mixing ratios at all eight layers, for both REF-C1 and REF-C2 simulations (in the case of REF-C1, the run ends in 2010, therefore we use the period of 1987-2010). The model data are interpolated to the vertical resolution of the ozonesonde, before being grouped into the same eight layers as defined before. Overall, the model captures the magnitude and variability of $O_3$ well, albeit with the overestimation of some observed peak values, mainly at the surface and in the stratosphere. The model is capable of reproducing the observed $O_3$ seasonal cycle in general, but underestimates the $O_3$ seasonal cycle in the free troposphere at this location. Comparing the two model experiments, i.e. REF-C1 and REF-C2, $O_3$ values in the REF-C1 experiment are generally larger than those in REF-C2, with the largest differences in the free troposphere at 3-6 km. Such differences might be due to the dynamical





differences arising from differences in sea surface temperatures between the two experiments. We calculate the linear trends of $O_3$ in both experiments, and in the sensitivity experiments. The results are discussed in the following sections.

## 4.2  REF-C1 simulation and the impact from $O_3$ precursor emissions and the stratosphere

Layer-resolved $O_3$ anomalies and their trends from REF-C1 and SEN-C1-fEMIS are displayed in Fig. 7 for the period of

1987-2010, together with the observed anomalies for the same period. In addition, the anomalies in the stratospheric $O_3$ tracer $O_3S$ are also displayed in the same plot. Note that modelled $O_3$ anomalies are also calculated based on monthly mean data, as applied to the observed anomalies. The modelled trends of $O_3$ and $O_3S$ are summarised in Table 4. Here, we do not expect that the model accurately captures the variability in $O_3$ anomalies, as the model is not entirely constrained by the observed meteorology (it is only driven by observed monthly-mean SSTs). Overall, the REF-C1 simulation captures some basic observed

variability but misses some large anomaly features, for example, the very negative $O_3$ anomalies near the surface in the year 2000, and at the beginning of the record (1987-1988). There are large observed variations in the upper troposphere (6-9 km) that the model fails to reproduce. The observed variabilities of $O_3$ around the tropopause and in the lower stratosphere are well captured by the model. The modelled trends match well with the observed trend in the tropopause region and in the upper troposphere (6-12 km), but the model cannot reproduce the observed significant positive $O_3$ trends below 6km. The model

simulation also produces a positive, but smaller than observed, trend in tropopause height ($7.7 \pm 1.9$ m/year) defined by the chemical tropopause of the 150 ppbv $O_3$ isopleth. Above 15km in the stratosphere, large positive anomalies from the model at the beginning of the record are contrary to the observed negative anomalies, and lead to the significant negative trends calculated in the REF-C1 simulation. In contrast, observed $O_3$ trends in the stratosphere are generally insignificant.

We consider the REF-C1 simulation to be a representation of the realistic $O_3$ evolution, driven by observed SSTs and well-

understood historic anthropogenic forcings such as GHGs and ODSs, including $CH_4$ which affects $O_3$ in both chemically and as a GHG. However, anthropogenic non-methane $O_3$ precursors have relatively large uncertainties, due to their very heterogeneous distributions and poorly understood emission inventories over some parts of the world. SEN-C1-fEMISS is designed to assess how such uncertainties in anthropogenic emissions of non-methane $O_3$ precursors might affect the $O_3$ evolution. We take the results of SEN-C1-fEMISS from 1987 to 2010, and display the anomaly and trend of $O_3$ in the same manner as $O_3$ from the

REF-C1 simulation (Fig. 7). The comparison shows that there is no apparent difference between $O_3$ trends in the troposphere between these two simulations, but there are distinct differences from the tropopause region to 25 km. However, the very high REF-C1 $O_3$ anomalies at the beginning of the record might contribute in part to the very negative trends in the stratosphere. The response of $O_3$ trend to changing non-methane $O_3$ precursors is quantified and shown in Table 5. Overall, the non-methane $O_3$ precursor emission changes between the late 1980s and 2010 have no significant effects on tropospheric $O_3$ trends, whereas

in the stratosphere, the simulation with constant emissions show flat or slightly negative trends in $O_3$, which indicates changes in emissions (both $NO_x$ and CO surface emissions show global increases during this period; Fig. 1, and Fig. 1 in Morgenstern et al. (2017)) contribute to the negative trends in stratospheric $O_3$.

The stratospheric tracer $O_3S$ depicted in Fig. 7 shows a weak but significant negative trend near the surface (Table 4). This decrease is likely the result of a decrease in stratospheric $O_3$ over this period. $O_3S$ is a measure of $O_3$ originating from the





stratosphere, and the negative trend in $O_3S$ also implies that the photochemical production of $O_3$ in the lower troposphere is increasing, most likely as a result of increasing $O_3$ precursors including methane. We mentioned before that there seems to be an increase in deep stratospheric intrusion at Lauder over the period of 1987-2014, as indicated by a negative trend in relative humidity. Clearly, STE plays an important role also in controlling lower tropospheric $O_3$, but a more rigorous attribution of

the trend in near-surface $O_3$ would require a more dedicated set of model experiments, which might be a worthwhile topic of future research.

### 4.3    REF-C2 simulation and the impact from ODSs, GHGs, and $CH_4$

In Fig. 8, we compare $O_3$ anomalies and their trends from the REF-C2 simulations to those from observations for the period of 1987-2014. The variations of observed $O_3$ anomalies are less accurately matched by the REF-C2 simulations, which are

free-running simulations not constrained by observed sea surface conditions, as opposed to the REF-C1 simulations which are driven by observed sea surface conditions. Significant negative trends in $O_3$ anomalies are calculated for the tropospheric layers, which are neither reflected in the observed trends nor in the REF-C1 simulations which yield a flat trend in the troposphere (Fig. 7). The significant negative tropospheric $O_3$ trends in REF-C2 are most likely the consequence of a sharp drop in surface $NO_x$ emissions after the year 2000 in the REF-C2 simulations, due to adopting a different emission dataset in 2000 (RCP6.0)

that differs from the MACCity emission inventory used in REF-C1 (see Fig. 1 and a related discussion in Morgenstern et al., 2017). This discrepancy in $NO_x$ emissions between REF-C1 and REF-C2 simulations results in significant differences in modelled $O_3$ trends at Lauder, a background SH mid-latitude site, indicating the important role of surface emissions of $O_3$ precursors impacting SH $O_3$ trends, most likely through inter-hemispheric transport. However, around the tropopause and in the stratosphere, the REF-C2 simulation reproduces the observed $O_3$ trends, in contrast to the REF-C1 simulations, although

the variations in observed $O_3$ are less well reproduced in REF-C2 than in REF-C1.

The purpose of using REF-C2 and SEN-C2 simulations is to assess the role of key forcing agents (ODSs, GHGs, and $CH_4$) in affecting the long term $O_3$ trend. In Fig. 8, we also display the $O_3$ trends from the sensitivity runs. For fixed-ODSs run (i.e. SEN-C2-fODS), the negative trends in $O_3$ are smaller in magnitudes than those from REF-C2 in the troposphere, which indicates that changes (i.e. increases) in ODSs between 1987 and 2015 would decrease the tropospheric $O_3$. However, there is

no significant difference in trends between REF-C2 and SEN-C2-fODS simulations in the stratosphere. Halogen loading in the atmosphere (Fig. 1) peaked in the 1990s and then dropped gradually approximately to the late-1980s level in 2014. This could explain the relatively small overall impact from ODSs over the period of 1987-2014 on $O_3$ at Lauder. The negative impact of ODSs changes during this period on tropospheric $O_3$ is also reflected in the $O_3S$ tracer which shows a small but significant negative trend over this period (Fig. 7).

Changes in GHGs over the period of 1987-2014 produce a negligible impact on simulated tropospheric $O_3$ changes at Lauder, but have significant impact on trends in stratospheric $O_3$ (above 12 km) (Fig. 8). The significant negative trends of $O_3$ in the fixed-GHGs simulations in the stratosphere imply increases in GHGs over 1987-2015 would contribute positively to the stratospheric $O_3$ trends. The effect is due to changes in $CO_2$, $CH_4$, and $N_2O$ collectively. We also performed a sensitivity simulation (SEN-C2-f$CH_4$) with only $CH_4$ fixed at the 1960 level, to separate the effect of $CH_4$ from other GHGs, e.g., $CO_2$





and $N_2O$. $CH_4$ is an $O_3$ precursor and also a GHG (see above). This analysis shows that in SEN-C2-fCH$_4$, significant negative trends in $O_3$ are calculated throughout the troposphere, as shown in Fig. 8, indicating that the increase in $CH_4$ over the period of 1987-2015 contributes positively to tropospheric $O_3$ trends. Given very similar $O_3$ trends in the REF-C2 and SEN-C2-fGHG simulations, it implies that increases in $CO_2$ and $N_2O$ would make negligible or slightly negative contributions to the $O_3$

trends over this period, out-weighing the positive contribution of increasing $CH_4$. In the stratosphere, $CH_4$ through changes in chemistry plays a much less significant role in $O_3$ trends.

Note that the period analysed (1987-2014) is relatively short, and the overall changes in ODSs and GHGs (including $CH_4$) are relatively small. Consequently, changes in $O_3$ trends attributable to these climate/chemical forcings are usually associated with large uncertainties , and therefore not significant at 95% confidence level (Table 5). Overall, changes in $CH_4$ contribute

positively to tropospheric $O_3$ trends through enhanced photochemical production of $O_3$. Increases in GHGs alone seem to result in decreasing tropospheric $O_3$ (presumably via enhanced chemical destruction in a future warmer and wetter climate) although this is counter balanced by increasing tropospheric $O_3$ through $CH_4$ oxidation. Increasing ODSs lead to significant negative $O_3$ trend in the troposphere, through downward transport of stratospheric $O_3$. In the stratosphere, the effect from ODSs is small due to the overall small changes in ODS over this period, whilst increases in GHGs seem to be the major forcing, contributing

positively to stratospheric $O_3$ trends through, possibly, decelerated $O_3$ destruction due to cooling. This positive contribution may have outweighed any reduction of stratospheric $O_3$ resulting from negative $O_3$ trends that is triggered by dynamical changes in the tropopause region, as both observations and REF-C2 simulations show relatively small and insignificant trends in stratospheric $O_3$ at Lauder. In the tropopause region, $O_3$ trends are not sensitive to changes in these forcings, and are largely controlled by dynamical changes, e.g., the movement of the tropopause. Both REF-C1 and REF-C2 produce a positive,

albeit smaller than observed, trend in tropopause height, which agrees with the scientific finding that anthropogenic forcings, in particular $O_3$ and well-mixed GHGs changes, contribute predominantly to observed tropopause height increase in recent decades (e.g., Santer et al., 2003).

In the following section, we expand the analysis to include the whole simulation period of 1960-2010, and assess long-term simulated changes in $O_3$ at SH mid-latitudes (represented by Lauder) due to changes in anthropogenic forcings of ODS, GHGs,

and $O_3$ precursor emissions.

### 4.4   Modelled attribution to long-term $O_3$ changes at Lauder

Figures. 9 and 10 display the $O_3$ evolution and trends over the 1960-2010 period from REF-C1 and REF-C2, respectively, and the associated sensitivity simulations. The $O_3$ anomalies are all normalised to zero at the starting time point. The analysis shows that in both simulations, $O_3$ shows insignificant or moderately negative trends in the troposphere, and more significant

negative trends in the stratosphere (Table 6).

Examining the sensitivity simulation SEN-C1-fEMIS, fixing emissions at the 1960 level results in negative trends throughout the domain, but with larger negative trends in the troposphere and smaller negative trends in the stratosphere, compared to REF-C1. This means that increases in $O_3$ precursor emissions since the 1960s lead to a positive contribution to tropospheric $O_3$ (shown in Fig. 9), predominantly through increased chemical $O_3$ production, but lead to a negative contribution to stratospheric





$O_3$. In the tropopause region (9-12 km), the impact on $O_3$ trends from fixed emissions is minimal, indicating changes in $O_3$ at the tropopause are mainly controlled by the movement of the tropopause rather than chemical changes. The trend in tropopause height in SEN-C1-fEMIS is very similar to that in REF-C1. In addition, the stratospheric tracer $O_3S$ from REF-C1 is also shown, which shows negative trends, as a result of the declining stratospheric $O_3$ during this period.

In the fixed-ODSs simulations, large differences in $O_3$ trends from the correspondent $O_3$ trends in REF-C2 are shown (Fig. 10), indicating significant negative contributions to $O_3$ trends due to changes (i.e. increases) in ODSs between 1960 and 2010. By comparison, changes in GHGs seem to have less impact on $O_3$ trends than ODSs changes. Increase in GHGs generally contribute positively to $O_3$ trends over this period, and the largest influence occurs in the troposphere, whereas the impact on stratospheric $O_3$ trends is negligible. Interestingly, the impact of changing GHGs on $O_3$ trends is shown mainly through $CH_4$

changes, suggesting that chemical changes ($O_3$ production through increased $CH_4$) are likely to dominate the contribution of changing GHGs to $O_3$ trends, especially given that the contribution mainly occurs in the troposphere (Fig. 10). We can not identify in this study the role of individual GHGs in moderating $O_3$ trends, and it is possible that radiative feedbacks are different from different GHGs, and chemical feedbacks also differ, e.g., $CH_4$ contributes to $O_3$ chemical production and $N_2O$ plays a role in stratospheric $O_3$ destruction. The attribution of modelled $O_3$ trends to each of the aforementioned forcings is

listed in Table 6.

## 5  Conclusions

We have analysed a 28-year ozonesonde record from Lauder, covering 1987 to 2014, and a surface $O_3$ record from Baring Head covering 1992 to 2015. Both background measurement sites are located in the SH mid-latitudes, and are representative of background atmospheric conditions. We have also analysed some meteorological parameters that are co-measured with $O_3$ at

Lauder, and have explored the relationships between $O_3$ changes and changes in these parameters, i.e., surface relative humidity, surface and stratospheric temperatures, and tropopause height, respectively. Through a regression analysis of ozonesonde data, which involves grouping the profiles into eight layers extending to the stratosphere (25 km), we have identified the dominant meteorological parameters that control the interannual variations of $O_3$ at Lauder. We find that relative humidity dominates the variability of lower tropospheric $O_3$, possibly through deep stratospheric intrusions, and that $O_3$ variability around the

tropopause region is dominated by the movement of the tropopause. In addition, ENSO contributes to $O_3$ variations in the free troposphere, and the QBO and solar cycle impact $O_3$ variations in the stratosphere.

The trends in observed $O_3$ at Lauder have been calculated for the eight layers. Significant positive trends in the troposphere below 6 km, and significant negative trends in the tropopause region are obtained, for both 1987-2010 and 1987-2014. No significant trends are found in the upper troposphere (6-9 km) and in the stratosphere (above 15 km). A significant negative $O_3$

trend in the 9-12 km region is very likely the result of an increasing tropopause height at Lauder. Such a dynamically induced change in $O_3$ also propagates to below and above the tropopause. Changes in temperatures in both the troposphere and the stratosphere impact $O_3$ by modifying the chemical production and destruction rates. Specifically, increased surface temperature would enhance photochemical production of $O_3$ mainly in the lower troposphere, and decreased stratospheric temperature





would slow down the nitrogen- and halogen-catalysed $O_3$ destruction cycles. Such effects from temperature changes could be balanced by the effect from tropopause height changes; the net impact are small $O_3$ trends in the upper troposphere and the lower stratosphere.

We have used NIWA-UKCA chemistry-climate model simulations to attribute $O_3$ trends observed at Lauder to anthropogenic
influences, particularly changing GHGs, ODSs, and $O_3$ precursors over 1987 to 2014. Results from these CCMI simulations indicate that $O_3$ precursor emissions have no net impact on tropospheric $O_3$, but lead to significant negative trends in strato- spheric $O_3$; this will be further investigated. Changes in $CH_4$ generally have a positive impact on $O_3$, but the effect is not significant at the 95% confidence level. Changes in GHGs (including $CH_4$) mainly affect the stratosphere where increased GHGs lead to significant positive trends in $O_3$. Increases in ODSs during this period mainly result in negative $O_3$ trend in the
troposphere, through stratosphere-troposphere exchange and/or consequences for tropospheric ozone photochemistry. But with $Br_y$ and $Cl_y$ peaking in the late 1990s, stratospheric $O_3$ at Lauder is not significantly affected by overall changes in ODSs from 1987 to 2014. We also examined $O_3$ transported from the stratosphere using a diagnostic stratospheric $O_3$ tracer, which shows a significant negative trend near the surface, suggesting that changes in stratosphere-troposphere exchange of ozone are indeed involved in tropospheric ozone trends.

The observed significant positive trend in tropospheric $O_3$ below 6 km is not reproduced by the base simulations (REF-C1 and REF-C2), which is very likely because of an underestimation of positive trends in surface emissions of $O_3$ precursors. The large significant negative trend in stratospheric $O_3$ in the REF-C1 simulation is caused by the very large positive anomaly at the beginning of $O_3$ record. Both base simulations reproduce well the significant negative $O_3$ trend in the tropopause region, associated with dynamical changes.

The Lauder $O_3$ record presented in this study is relatively short for detecting significant responses of $O_3$ to changes in GHGs, ODSs, and $O_3$ precursors. The analysis is extended to cover the whole period of the simulation, i.e., 1960-2010, during which period significant changes in GHGs, ODSs, and $O_3$ precursors occur, allowing for a more robust assessment of their impacts on the $O_3$ evolution. The results show that all forcings contribute significantly to simulated $O_3$ trends over the last five decades at Lauder. Increases in $O_3$ precursors contribute positively to tropospheric $O_3$ through enhanced photochemical production of $O_3$,
but negatively to stratospheric $O_3$. The increase in $CH_4$ leads to positive $O_3$ trends in both the troposphere and the stratosphere, and is comparable to the $O_3$ response to the net effect of combined GHGs changes, indicating there may be a cancellation of effects from increases in $CO_2$ and in $N_2O$. The overall increase in ODSs between 1960 and 2010 results in significant negative trend in tropospheric and stratospheric $O_3$, which coincides with the negative trend in the stratospheric $O_3$ tracer.

This study demonstrates that long-term ozone profile observations provide valuable insight into atmospheric composition
changes associated with anthropogenic forcing over the recent decades at a background SH mid-latitude location. More loca- tions can be explored in future studies and multi-model simulation ensembles can be used to derive global $O_3$ changes, and to project the future evolution.



*Data availability.* All data used in this paper can be obtained from the contact author. The Lauder ozonesonde data are available at ftp://ftp.cpc.ncep.noaa.gov/ndacc/station/lauder/ames/o3sonde/. Baring Head surface $O_3$ data are available at http://ds.data.jma.go.jp/gmd/wdcgg. NIWA-UKCA simulations can be downloaded from the CCMI archive; see instructions at http://blogs.reading.ac.uk/ccmi/badc-data-access/

*Competing interests.* The authors declare that they have no conflict of interest.

*Acknowledgements.* We acknowledge the NIWA Lauder Ozone team for ozone sonde measurements since 1986, and technicians involved at various stages of data collection. We acknowledge the U.K. Met Office for use of the MetUM. Furthermore, we acknowledge the contribution of NeSI high-performance computing facilities to the results of this research. NZ's national facilities are provided by the NZ eScience Infrastructure and funded jointly by NeSI's collaborator institutions and through the Ministry of Business, Innovation & Employment's Research Infrastructure programme (https://www.nesi.org.nz). This research was supported by the NZ Government's Strategic Science Investment Fund (SSIF) through the NIWA programmes CACV and CAAC, and by the Marsden Fund Council from New Zealand Government funding, managed by the Royal Society of New Zealand, under project 12NIW006.





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



**Table 1.** Chemistry-Climate Model Initiative (CCMI) simulations performed by NIWA-UKCA, and used in this study.

| Simulation | Period | SSTs/sea ice | $O_3$ precursor Emissions | $CH_4$ | GHGs | ODSs |
|---|---|---|---|---|---|---|
| REF-C1 | 1960-2010 | HADISST | MACCity (1960-2010) | RCP 6.0 | RCP 6.0 | WMO (2014) A1 |
| SEN-C1-fEMIS | 1960-2010 | HADISST | Fixed at 1960 | RCP 6.0 | RCP 6.0 | WMO (2014) A1 |
| REF-C2 | 1960-2100 | Interactive | MACCity (1960-2000) & RCP 6.0 (2000-2100) | RCP 6.0 | RCP 6.0 | WMO (2014) A1 |
| SEN-C2-fGHG | 1960-2100 | Interactive | MACCity (1960-2000) & RCP 6.0 (2000-2100) | RCP 6.0 | Fixed at 1960 | WMO (2014) A1 |
| SEN-C2-fCH$_4$ | 1960-2100 | Interactive | MACCity (1960-2000) & RCP 6.0 (2000-2100) | Fixed at 1960 | RCP 6.0 | WMO (2014) A1 |
| SEN-C2-fODS | 1960-2100 | Interactive | MACCity (1960-2000) & RCP 6.0 (2000-2100) | RCP 6.0 | RCP 6.0 | Fixed at 1960 |

**Table 2.** Dominant contributors to the regression, in order of importance, resolved by altitude.

| Altitudes (km) | Dominant regression functions |
|---|---|
| 0-1.5 | Relative humidity at the surface |
| 1.5-3 | Tropopause height; Relative humidity at the surface |
| 3-6 | Tropopause height; Stratospheric T; ENSO |
| 6-9 | Tropopause height; Stratospheric T; ENSO |
| 9-12 | Tropopause height |
| 12-15 | Tropopause height |
| 15-20 | Tropopause height; QBO at 30 hPa; Stratospheric T; Solar cycle |
| 20-25 | Stratospheric T; QBOs at 30 hPa and at 10 hPa |




**Table 3.** Correlation coefficients between $O_3$ anomalies and dominant climate/meteorological variables at each layer. Missing entries indicate absolute correlation coefficients less than 0.1.

| Ozone layers | RH | $\mathbf{T}_{surf}$ | $\mathbf{T}_{strat}$ | MEI | QBO30 |
|---|---|---|---|---|---|
| $\mathbf{O}_3$ (0-1.5km) | $-0.50$ | 0.38 | | | $-0.12$ |
| $\mathbf{O}_3$ (1.5-3km) | $-0.28$ | 0.35 | $-0.17$ | $-0.11$ | |
| $\mathbf{O}_3$ (3-6km) | $-0.20$ | 0.42 | $-0.29$ | $-0.18$ | |
| $\mathbf{O}_3$ (6-9km) | | | | $-0.15$ | |
| $\mathbf{O}_3$ (9-12km) | 0.21 | $-0.44$ | 0.31 | | |
| $\mathbf{O}_3$ (12-15km) | 0.23 | $-0.45$ | 0.27 | 0.13 | |
| $\mathbf{O}_3$ (15-20km) | | $-0.36$ | 0.32 | 0.10 | $-0.32$ |
| $\mathbf{O}_3$ (20-25km) | | $-0.18$ | 0.42 | | $-0.31$ |





**Table 4.** Observed and simulated trends in $O_3$ and $O_3S$ anomalies ($\pm 1$ Standard Error) at Lauder, over the periods of 1987-2010 and 1987-2014, accordingly.

| Altitute | Observed $O_3$ | Observed $O_3$ | REF-C1 $O_3$ | REF-C2 $O_3$ | REF-C1 $O_3S$ |
|---|---|---|---|---|---|
| km | ppbv yr$^{-1}$ | | | | |
| | 1987-2010 | 1987-2014 | 1987-2010 | 1987-2014 | 1987-2010 |
| 0-1.5 | $0.06 \pm 0.02$ | $0.04 \pm 0.01$ | $0.00 \pm 0.01$ | $-0.05 \pm 0.01$ | $-0.02 \pm 0.00$ |
| 1.5-3 | $0.08 \pm 0.02$ | $0.04 \pm 0.01$ | $0.01 \pm 0.01$ | $-0.06 \pm 0.01$ | $-0.02 \pm 0.01$ |
| 3-6 | $0.06 \pm 0.03$ | $0.05 \pm 0.02$ | $0.02 \pm 0.02$ | $-0.07 \pm 0.01$ | $0.00 \pm 0.01$ |
| 6-9 | $-0.09 \pm 0.05$ | $-0.01 \pm 0.05$ | $-0.01 \pm 0.02$ | $-0.10 \pm 0.02$ | $-0.04 \pm 0.02$ |
| 9-12 | $-0.55 \pm 0.27$ | $-0.62 \pm 0.21$ | $-0.39 \pm 0.12$ | $-0.26 \pm 0.07$ | $-0.47 \pm 0.13$ |
| 12-15 | $-0.53 \pm 0.44$ | $-0.90 \pm 0.35$ | $-1.18 \pm 0.32$ | $-0.51 \pm 0.23$ | $-1.19 \pm 0.32$ |
| 15-20 | $0.89 \pm 1.12$ | $-1.01 \pm 0.88$ | $-4.19 \pm 0.73$ | $-1.00 \pm 0.56$ | $-4.19 \pm 0.73$ |
| 20-25 | $1.46 \pm 1.61$ | $-0.50 \pm 1.28$ | $-4.46 \pm 0.95$ | $0.20 \pm 0.69$ | $-4.56 \pm 0.96$ |

**Table 5.** Attributions of modelled trends ($\pm 1$ Standard Error) due to changes in non-methane $O_3$ precursors, and in $CH_4$, GHGs, and ODS over the period of 1987-2010 and 1987-2014, respectively, calculated as the trends in absolute differences between $O_3$ anomalies from the sensitivity simulations and $O_3$ anomalies from the REF-C1 and REF-C2 simulation, respectively.

| Altitude | Due to Emissions | Due to CH$_4$ | Due to GHGs | Due to ODSs |
|---|---|---|---|---|
| km | ppbv yr$^{-1}$ | | | |
| | 1987-2010 | 1987-2014 | 1987-2014 | 1987-2014 |
| 0-1.5 | $-0.00 \pm 0.01$ | $0.02 \pm 0.02$ | $-0.00 \pm 0.01$ | $-0.02 \pm 0.01$ |
| 1.5-3 | $-0.00 \pm 0.02$ | $0.02 \pm 0.02$ | $-0.00 \pm 0.01$ | $-0.03 \pm 0.02$ |
| 3-6 | $0.02 \pm 0.02$ | $0.02 \pm 0.02$ | $-0.01 \pm 0.02$ | $-0.05 \pm 0.02$ |
| 6-9 | $0.00 \pm 0.03$ | $0.00 \pm 0.03$ | $0.00 \pm 0.02$ | $-0.08 \pm 0.03$ |
| 9-12 | $-0.42 \pm 0.16$ | $-0.05 \pm 0.11$ | $0.20 \pm 0.11$ | $0.02 \pm 0.12$ |
| 12-15 | $-1.05 \pm 0.40$ | $0.23 \pm 0.51$ | $0.69 \pm 0.33$ | $0.14 \pm 0.37$ |
| 15-20 | $-2.49 \pm 0.91$ | $0.50 \pm 1.56$ | $1.27 \pm 0.79$ | $-0.11 \pm 0.92$ |
| 20-25 | $-3.20 \pm 1.24$ | $1.80 \pm 1.99$ | $3.91 \pm 1.01$ | $0.52 \pm 1.09$ |





**Table 6.** Simulated trends in $O_3$ anomalies ($\pm 1$ Standard Error) at Lauder, and attributions of modelled trends due to changes in non-methane $O_3$ precursors, $CH_4$, GHGs, and ODS over the period of 1960-2010.

| Alt. | REF-C1 | REF-C2 | Due to Emiss. | Due to CH$_4$ | Due to GHGs | Due to ODSs | O$_3$S (REF-C1) |
|---|---|---|---|---|---|---|---|
| km | \multicolumn ppbv yr$^{-1}$ | | | | | | |
| 0-1.5 | $0.00 \pm 0.00$ | $-0.02 \pm 0.00$ | $0.02 \pm 0.01$ | $0.03 \pm 0.01$ | $0.03 \pm 0.01$ | $-0.07 \pm 0.01$ | $-0.04 \pm 0.00$ |
| 1.5-3 | $0.00 \pm 0.00$ | $-0.02 \pm 0.00$ | $0.02 \pm 0.01$ | $0.04 \pm 0.01$ | $0.04 \pm 0.01$ | $-0.09 \pm 0.01$ | $-0.05 \pm 0.00$ |
| 3-6 | $0.00 \pm 0.01$ | $-0.03 \pm 0.01$ | $0.03 \pm 0.01$ | $0.05 \pm 0.01$ | $0.05 \pm 0.01$ | $-0.11 \pm 0.01$ | $-0.05 \pm 0.00$ |
| 6-9 | $-0.05 \pm 0.01$ | $-0.04 \pm 0.01$ | $0.04 \pm 0.01$ | $0.05 \pm 0.01$ | $0.06 \pm 0.01$ | $-0.16 \pm 0.01$ | $-0.12 \pm 0.01$ |
| 9-12 | $-0.51 \pm 0.05$ | $-0.26 \pm 0.04$ | $-0.26 \pm 0.06$ | $0.06 \pm 0.05$ | $0.05 \pm 0.05$ | $-0.49 \pm 0.05$ | $-0.60 \pm 0.05$ |
| 12-15 | $-1.33 \pm 0.13$ | $-0.80 \pm 0.11$ | $-0.29 \pm 0.15$ | $0.33 \pm 0.16$ | $0.29 \pm 0.16$ | $-1.24 \pm 0.16$ | $-1.33 \pm 0.13$ |
| 15-20 | $-5.12 \pm 0.26$ | $-3.95 \pm 0.26$ | $-1.68 \pm 0.33$ | $0.49 \pm 0.39$ | $0.42 \pm 0.38$ | $-4.05 \pm 0.38$ | $-5.12 \pm 0.26$ |
| 20-25 | $-7.88 \pm 0.31$ | $-6.25 \pm 0.34$ | $-2.51 \pm 0.41$ | $0.81 \pm 0.48$ | $0.77 \pm 0.45$ | $-6.93 \pm 0.47$ | $-7.88 \pm 0.31$ |



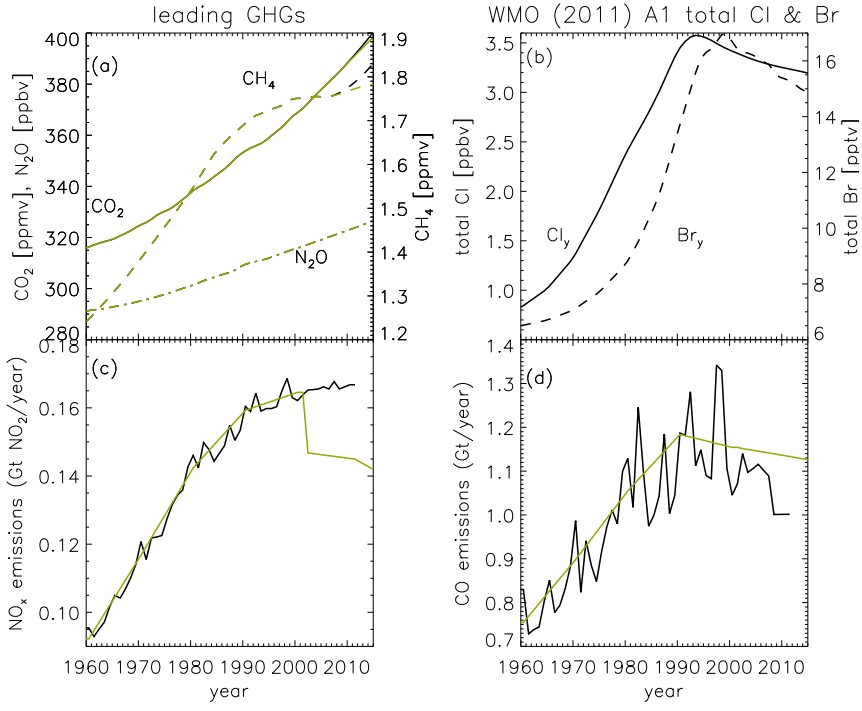

**Figure 1.** Time evolution of anthropogenic forcings used in CCMI simulations: (a) greenhouse gases (GHGs), (b) ozone depleting substances (ODSs), (c) nitrogen oxides ($NO_x$), and (d) carbon monoxide (CO). Black lines denote forcings used in REF-C1, and lime lines for REF-C2.





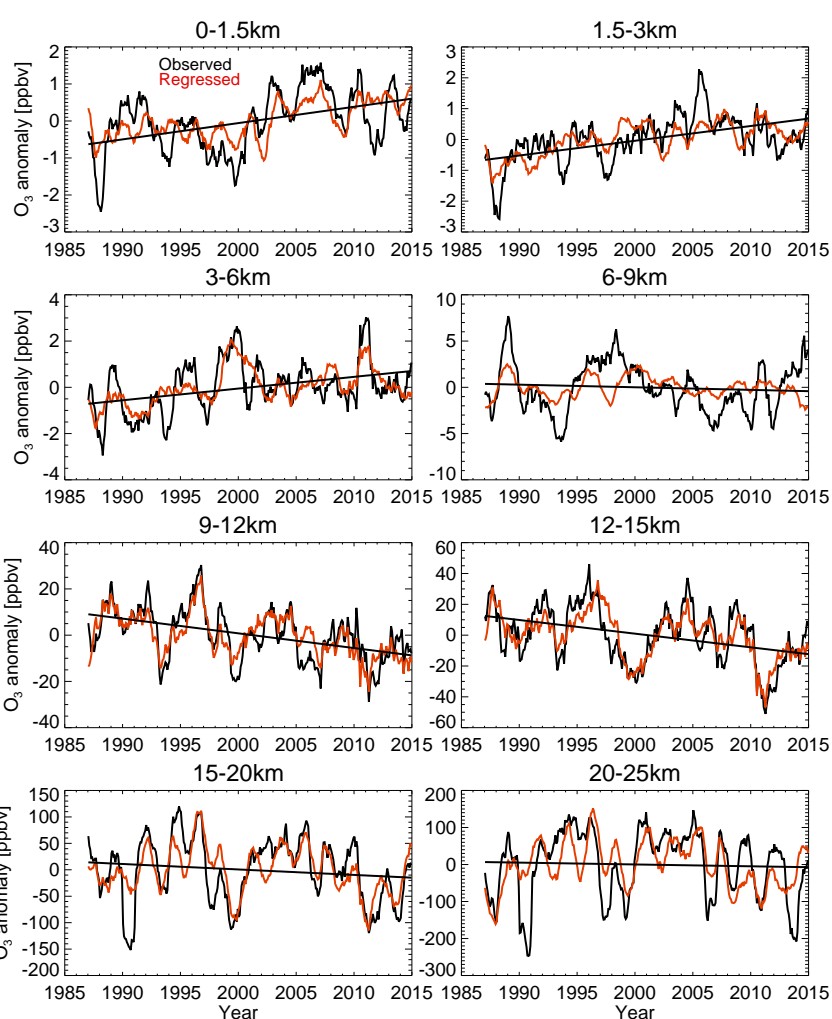

**Figure 2.** Observed ozone anomalies and trends at Lauder (1987-2014), and O$_3$ anomalies fitted with regression functions. Data are monthly mean and smoothed using a 13-month filter.





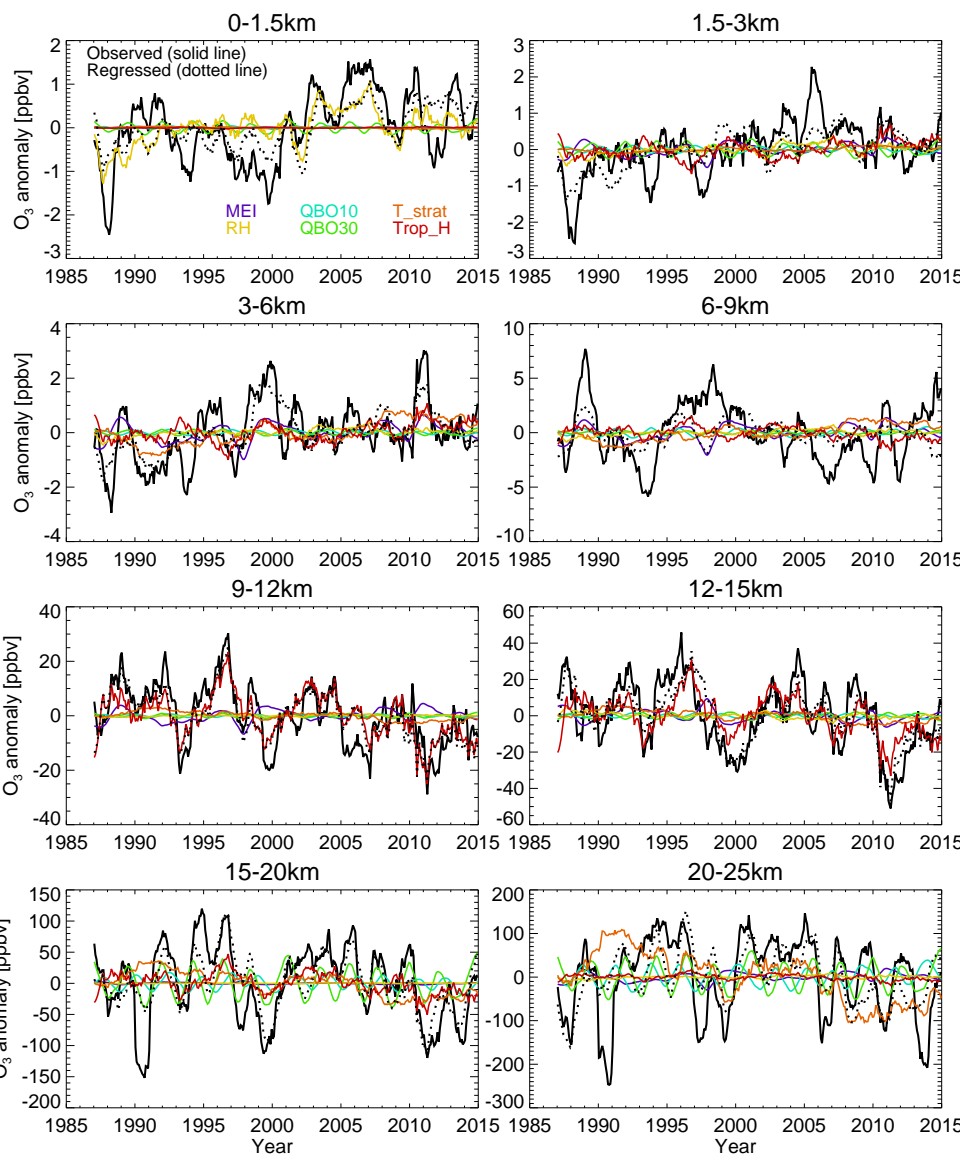

**Figure 3.** Ozone anomalies and contributions of individual regression functions at Lauder. Data are monthly mean and smoothed using a 13-month filter.





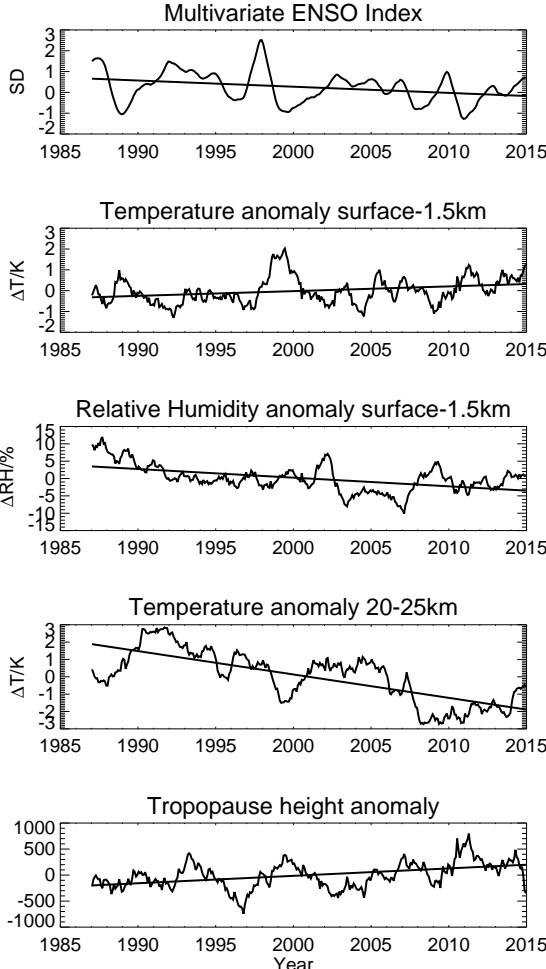

**Figure 4.** Time series of dominant climate/meteorological variables, co-measured with ozonesonde at Lauder, with the exception of the ENSO Index. Data are monthly mean and smoothed using a 13-month filter.





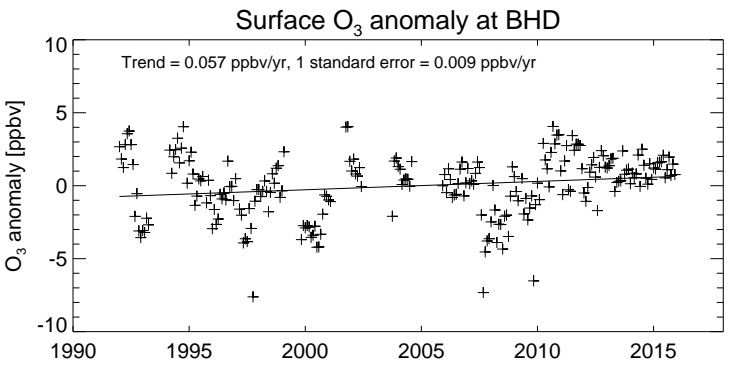

**Figure 5.** Observed monthly mean surface $O_3$ anomaly and trend at Baring Head (1994-2015).



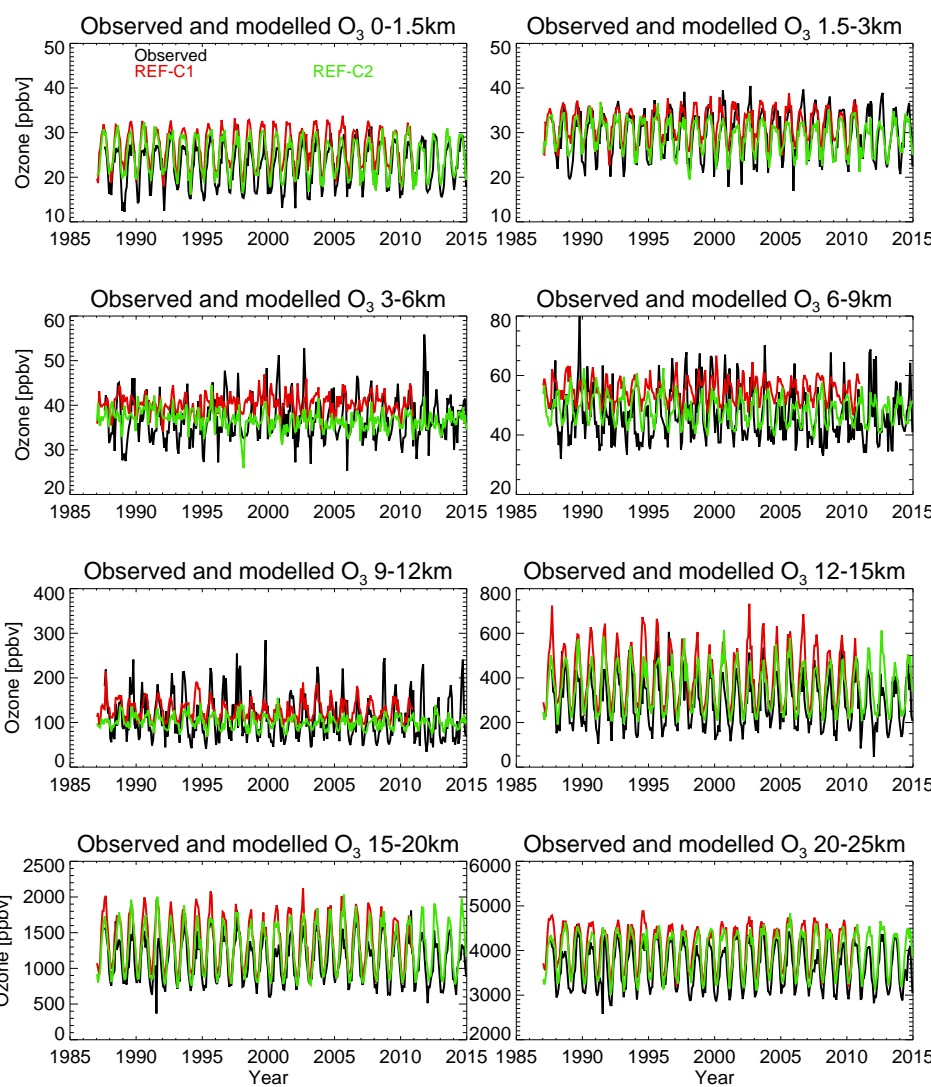

**Figure 6.** Observed and modelled ozone time series at Lauder over 1987-2014.





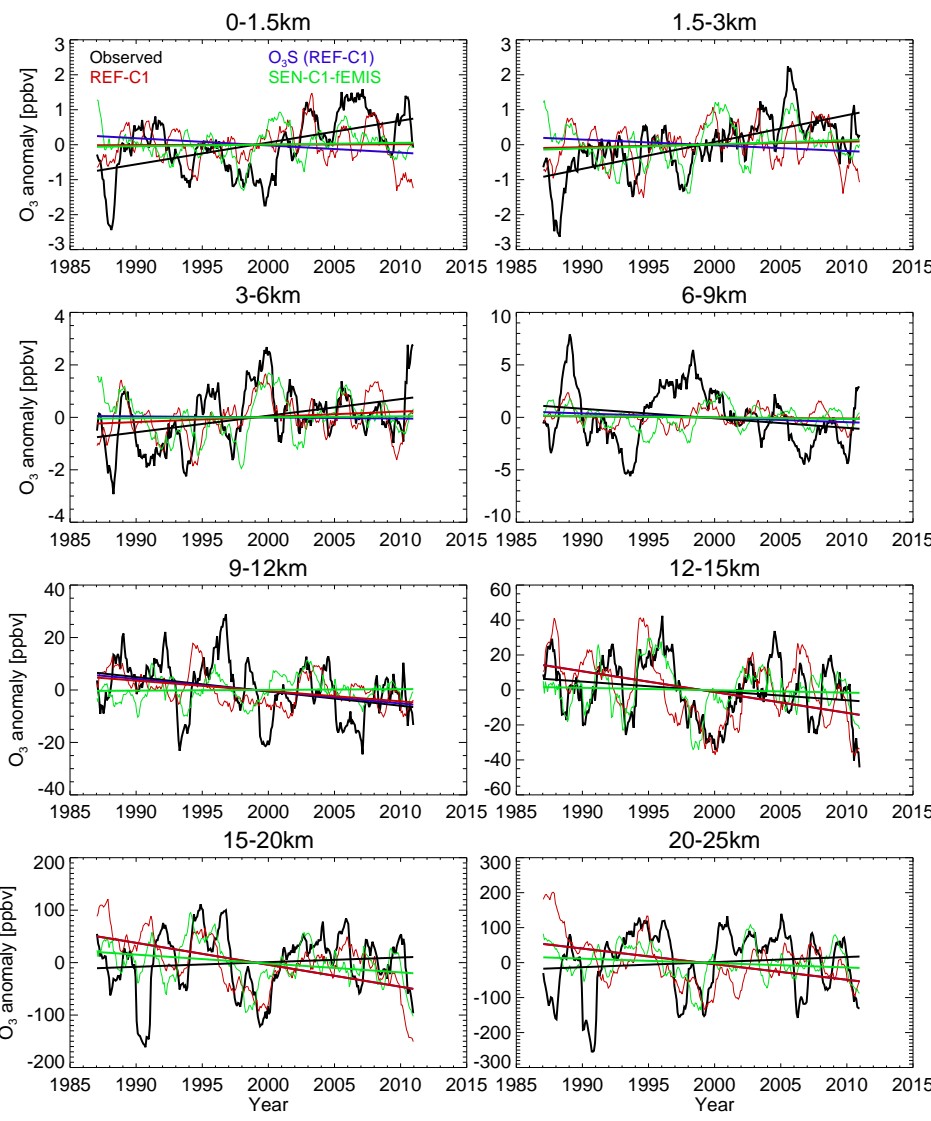

**Figure 7.** Observed and modelled O₃ anomaly and trend at Lauder (1987-2010). Model results are from REF-C1 and SEN-C1-fEMIS simulations, respectively.



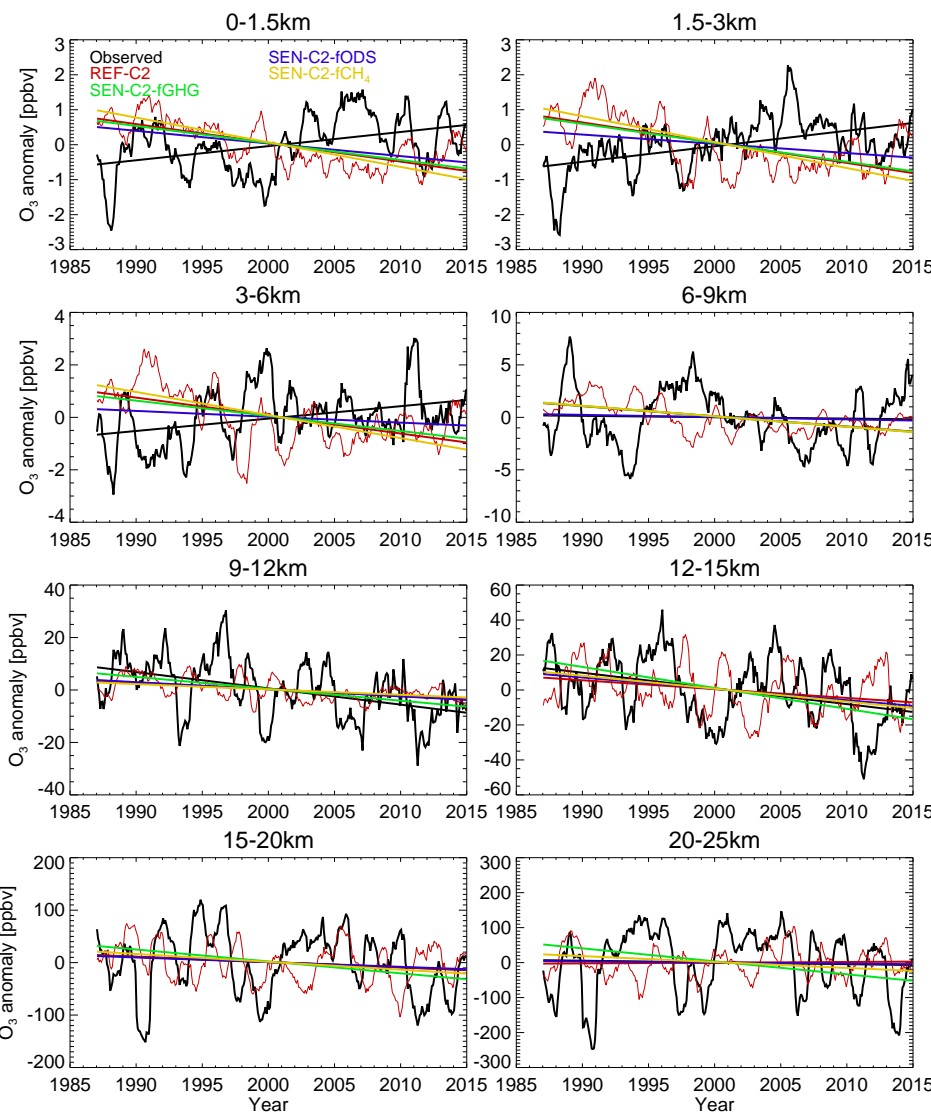

**Figure 8.** Observed and modelled (REF-C2) $O_3$ anomaly and trend at Lauder over 1987-2014. Modelled trend from SEN-C2-fGHG, SEN-C2-fCH$_4$, and SEN-C2-fODS are also depicted.




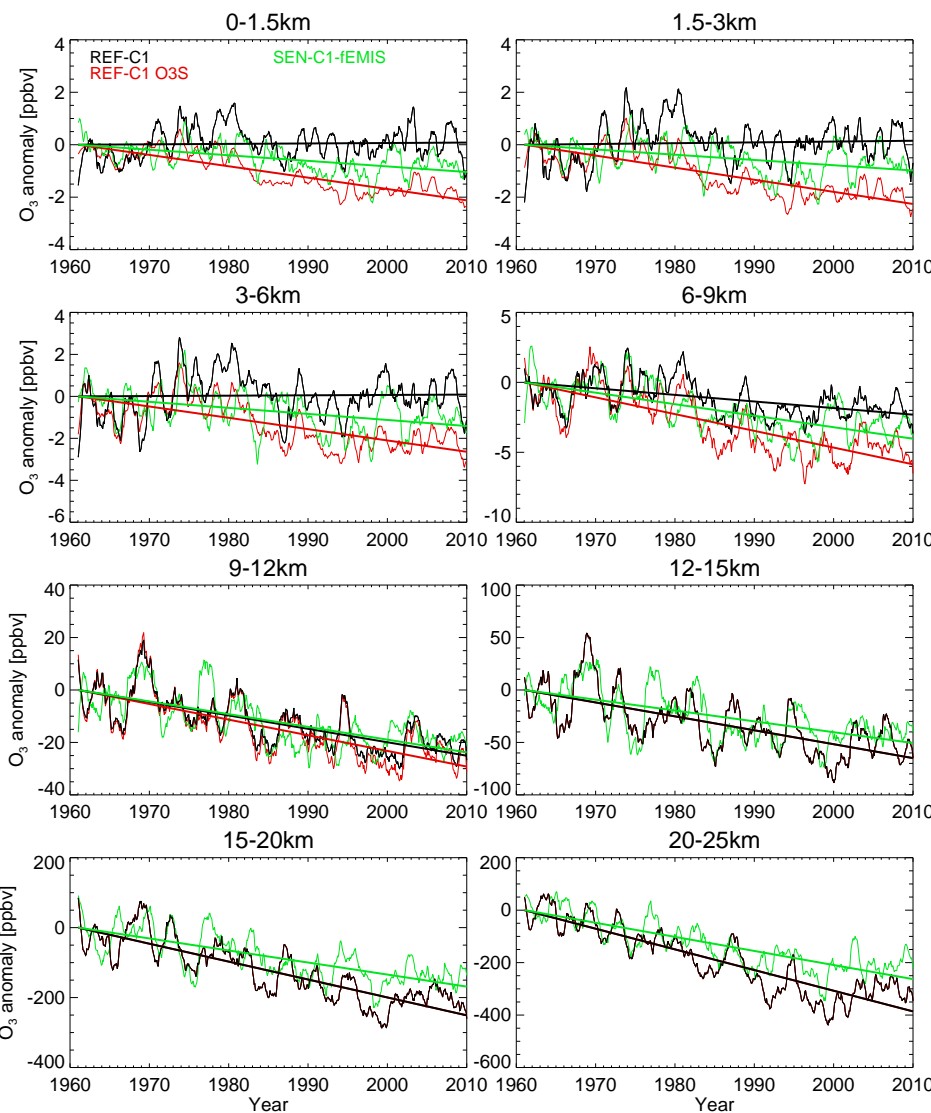

**Figure 9.** Modelled O$_3$ anomaly and trend at Lauder over 1960-2010 from REF-C1 and SEN-C1-fEMIS, respectively. The stratospheric O$_3$ tracer is also depicted.





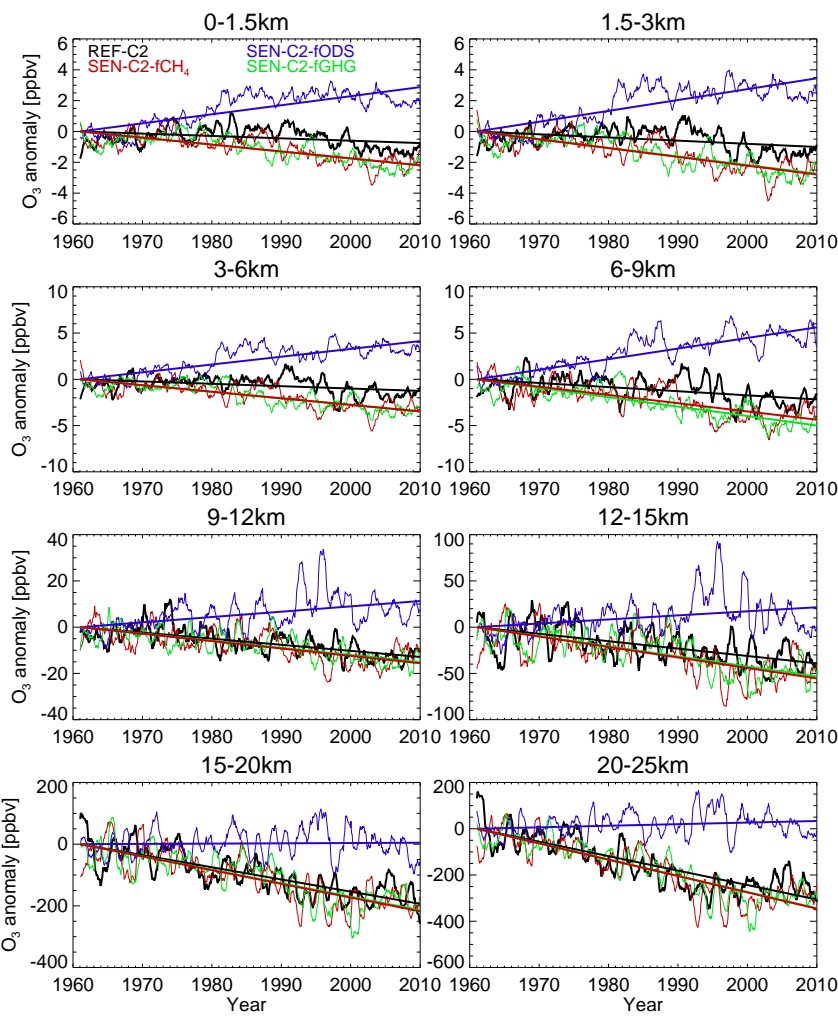

**Figure 10.** Modelled $O_3$ anomalies and trends at Lauder over 1960-2010, from REF-C2, SEN-C2-fGHG, SEN-C2-fCH$_4$, and SEN-C2-fODS, respectively.