# Peer review of "Attribution of recent ozone changes in the Southern Hemisphere mid-latitudes using statistical analysis and chemistry-climate model simulations"

_Atmospheric Chemistry and Physics, 2017_

## Referee Comment (RC1) · Anonymous Referee #1 · 22 May 2017

This paper presents an interesting analysis of ozone variability above the observation site of Lauder in New Zealand. The objective is to identify and quantify the main drivers of ozone variability and trend at different altitudes as monitored with ozone sondes. The attribution is carried out using multivariate regression analysis and sensitivity simulations from a chemistry-climate model. A large part of the ozone variability is found to be driven by dynamical/climate variability. Some of it is also linked to changes in O3 precursors emissions. The results suggest that ozone long-term monitoring at specific sites contains valuable information in terms of the causes of ozone changes. This study illustrates how to extract information about atmospheric composition changes from observational time series. The results will be useful to scientists running monitoring sites. The paper is reasonably clear and well written. Its scope fits perfectly with those of ACP. Therefore, I recommend publication with minor corrections that the authors may consider.

l19, p2 : 'where climate change accelerates the export of O3 to higher latitudes, reducing O3' I think the impact of climate change on tropical ozone in the stratosphere is more about enhanced tropical uplift and hence reduced time for ozone production in rising air.

l5, p3: for more clarity, I would suggest to have: 2.1 ozone record 2.2 statistical method 2.3 chemistry-climate model simulations

l21, p3: 'Tropopause height is calculated using the 150 ppbv O3 chemical tropopause'. Why 150 pbbv ? Prather et al., 2011, recommended the 100 ppbv O3 contour. Prather, M. J., et al. (2011), An atmospheric chemist in search of the tropopause, J. Geophys. Res., 116, D04306, doi:10.1029/2010JD014939.

l24, p3: The different model simulations are forced by different scenarios in CH4 and other O3 precursors, greenhouse gases (GHGs), and ozone depleting substances (ODSs) over the period of the time series. There does not seem to be any forcing that is characteristic of the effect of tropospheric O3 precursors.

l24, p3: The effect of surface temperature is discussed in the text and appears in Table 3 (correlation coefficient). It seems to be the dominant factor in O3 variability at the surface. Why wasn't it included in the regression?

l20, p3: What is Cly? 'effective equivalent chlorine loading (Cly )'. and 2 lines down : 'Cly is the total chlorine loading'.

l27, p3 : 'We note that interdependencies and correlations, e.g., between stratospheric temperature, relative humidity at surface, and tropopause height as regression functions, cannot be excluded'. There is no need to speculate here. For example, the effective equivalent chlorine loading, a regression function, impacts O3 and stratospheric temperature, one of the regression function. Furthermore, correlations between regression functions can be calculated. It is a significant source of uncertainties and should be discussed.

l24, p4: 'CH4 mixing ratios are prescribed at the surface, and the same CH4 scenario is used in both chemistry and radiation.' The wording is a bit confusing. Isn't the model-calculated CH4 fed into the radiation scheme, like O3?

l32, p4: This paragraph would greatly gain in clarity and usefulness if the different simulations were described and contrasted instead of just referring to the Table 1.

l10, p5: 'Fig. 2 shows the deseasonalised O3 anomalies at the eight layers from the surface to the lower stratosphere, and the respective regressed O3 anomalies'. Please, could you indicate when a figure is about observed or model-calculated variables? Also, is it monthly means for all the analysis (as indicated in one figure)? Give more information about the time resolution and data processing/filtering of the different analyses.

l11, p5: Can be quantitative about the amount of variability captured by the regression? just provide the values of the determination coefficient R2.

l19, p5: "project onto" ? replace by "are mostly driven by"

l26, p5: se comment above about Tsurf missing from the regression.

l29, p6: it's very unlikely that temperature is the driver of O3 variability in the lower stratosphere. The slowing down of O3 destruction by stratospheric cooling (via Chapman cycle) occurs in the upper stratosphere. The ozone budget and variability in the mid-latitude lower stratosphere is dominating by dynamics. I think temperature changes simply reflect dynamical changes that drive O3 variability.

l9, p8: Difficult to expect a CCM to reproduce specific short-term O3 anomalies. Those anomalies are often driven by specific dynamical variations. Comparisons between

CCM simulations and observational time series make more sense when considering long term trends. I would suggest to add the analysis of a REF-C1SD simulation (wind/temperature forced by meteorological analyses instead of being calculated by the model) and compare to REF-C1. It would give an estimate about the effect of model-calculated dynamics/meteorology and biases (yes, there are some) on O3 variability, including trend, for the considered site. Even on long-time series, this could be significant. The authors would be in a stronger position in their attribution analysis. It is less of a problem when considering large-scale averages or multiple sites but here the analysis is limited to a specific site.

Figures : Can the authors add in all the figure captions which curves are what?

---

## Referee Comment (RC2) · Anonymous Referee #2 · 26 May 2017

Review of Zeng et al. Attribution of recent ozone changes in the Southern Hemisphere mid-latitudes using statistical analysis and chemistry-climate model simulations

This study uses statistical analysis to understand recent changes in ozone up to 25 km at Lauder, NZ and relate them to changes over similar and longer time periods in chemistry-climate model simulations. Overall the manuscript is clearly written and analysis explained well and I think it would be of interest to the ACP community and would recommend its publication after the authors take some mostly minor comments into consideration.

[Figure]

General comments:

I have concerns about using a chemical measure (O3) for tropopause height to evaluate the O3 field itself. Can you demonstrate that this measure is the same as a PV or temperature based tropopause height over varying timescales?

I can understand on shorter timescales how the level of o3 of 150 ppb would reflect dynamical variability in tropopause height (although one could argue that a different concentration ∼100 ppb might be a better choice) but on longer timescales it seems like the chemical changes (through ODS changes) would also be reflected in this quantity. Can you quantify and separate and include some discussion in the text of this issue?

What does chemical O3 tropopause changes look like over the full ref-C2 run can you see the impact of ODS changes causing o3 loss and recovery reflected in this quantity? If you can't see this impact it would be a good demonstration that it is essentially a dynamics only representation over both short and long time-scales. If you do it can be quantified and related to the smaller change in ODSs over the Lauder record

I understand that the Lauder record 1987-2014 includes only a modest change in ODSs peaking in the 1990s with little net change so a linear trend might be expected to be flat but it doesn't rule out a some ozone loss in the early period and gain in the later period.

Can you explain why the variables in Table 3 are in some cases different than those shown in Figure 3 like surface Temperature which seems to be highly correlated to lower troposphere ozone is not shown in figure 3.

p5 line 17-18 it seems a bit circular argument if you are using o3 to define the tropopause height.

p10 line1-3 Can you make the same claim for ODS changes over this time refC1 time period (difference between RefC1 and fODS).

p10 line 34 p11 line 1-2 it is not obvious why this should be the case, can you do any

additional analysis to explain possible mechanisms.

---

## Author Comment (AC1) · 28 Jul 2017

This paper presents an interesting analysis of ozone variability above the observation site of Lauder in New Zealand. The objective is to identify and quantify the main drivers of ozone variability and trend at different altitudes as monitored with ozone sondes. The attribution is carried out using multivariate regression analysis and sensitivity simulations from a chemistry-climate model. A large part of the ozone variability is found to be driven by dynamical/climate variability. Some of it is also linked to changes in O3 precursors emissions. The results suggest that ozone long-term monitoring at specific sites contains valuable information in terms of the causes of ozone changes. This study illustrates how to extract information about atmospheric composition changes from observational time series. The results will be useful to scientists running monitoring sites. The paper is reasonably clear and well written. Its scope fits perfectly with those of ACP. Therefore, I recommend publication with minor corrections that the authors may consider.

**Thanks very much for the reviewer's positive comments.**

l19, p2 : 'where climate change accelerates the export of O3 to higher latitudes, reducing O3' I think the impact of climate change on tropical ozone in the stratosphere is more about enhanced tropical uplift and hence reduced time for ozone production in rising air.

**We have revised this sentence to "…but there may be no sustained recovery in the tropical lower stratosphere where climate change increases tropical upwelling, leading to less time for O$_3$ production and hence reducing O$_3$ in this region (Eyring et al., 2010)." (Page 2, lines 18-20)**

l5, p3: for more clarity, I would suggest to have: 2.1 ozone record 2.2 statistical method 2.3 chemistry-climate model simulations

**Thanks for the suggestion. We have now added subsections in Section 2.**

l21, p3: 'Tropopause height is calculated using the 150 ppbv O3 chemical tropopause'. Why 150 pbbv ? Prather et al., 2011, recommended the 100 ppbv O3 contour. Prather, M. J., et al. (2011), An atmospheric chemist in search of the tropopause, J. Geophys. Res., 116, D04306, doi:10.1029/2010JD014939.

**We have now adopted the definition of the tropopause based on vertical gradient of ozone, which is greater than 60 ppbv/km and remains so for a further 200 meters, constrained by ozone mixing ratio greater than 80 ppbv and exceeds 110 ppbv immediate above the tropopause (Bethan et al., 1996). This ozone tropopause definition has the advantage of being consistent with the PV definition (Beekmann et al., 1994, Bethan et al., 1996)). Consequently, we have recalculated the regression function and the trend of the observed and simulated tropopause height based on revised tropopause height, and revised figures (2, 3, and 4), tables (2 and 3), and the relevant text accordingly. Resultant changes are small. (Page 3, line 24-28)**

l24, p3: The different model simulations are forced by different scenarios in CH4 and other O3 precursors, greenhouse gases (GHGs), and ozone depleting substances (ODSs) over the period of the time series. There does not seem to be any forcing that is characteristic of the effect of tropospheric O3 precursors.

**We do not include short-lived ozone precursors in the regression as this would be difficult to formulate in terms of a usable, univariate regression function. Moreover, CH$_4$ is the most effective ozone precursor in the remote Southern Hemisphere background air, but it doesn't impact ozone variability due to its well-mixed nature.**

l24, p3: The effect of surface temperature is discussed in the text and appears in Table 3 (correlation coefficient). It seems to be the dominant factor in O3 variability at the surface. Why wasn't it included in the regression?

**We have now included surface temperature as a regression function, and made changes accordingly. As a result, we have revised Figures 2 and 3, and Tables 2 and 3 accordingly, and have added related discussions in Section 3.1. (Page 3, Eqn. (1) and Sec. 3.1)**

l20, p3: What is Cly? 'effective equivalent chlorine loading (Cly )'. and 2 lines down : 'Cly is the total chlorine loading'.

**Thank you. We have now clarified that "Cly is the total chlorine loading". (Page 3, line 23)**

l27, p3 : 'We note that interdependencies and correlations, e.g., between stratospheric temperature, relative humidity at surface, and tropopause height as regression functions, cannot be excluded'. There is no need to speculate here. For example, the effective equivalent chlorine loading, a regression function, impacts O3 and stratospheric temperature, one of the regression function. Furthermore, correlations between regression functions can be calculated. It is a significant source of uncertainties and should be discussed.

**We have removed this sentence and have added a discussion of the correlations between regression functions in Section 3.1. (Page 6, lines 32-34 & page 7, lines 1-2)**

l24, p4: 'CH4 mixing ratios are prescribed at the surface, and the same CH4 scenario is used in both chemistry and radiation.' The wording is a bit confusing. Isn't the model-calculated CH4 fed into the radiation scheme, like O3?

**Yes, model-calculated $CH_4$ (i.e. prescribed at the surface but subjected to transport and chemical removal) feeds into the radiation scheme. We have revised the text to make this clearer. (Page 5, lines 2-3)**

l32, p4: This paragraph would greatly gain in clarity and usefulness if the different simulations were described and contrasted instead of just referring to the Table 1.

**We have now modified the texts to make the description of this second set of CCMI simulations clearer. (Page 5, lines 10-12, 16-18)**

l10, p5: 'Fig. 2 shows the deseasonalised O3 anomalies at the eight layers from the surface to the lower stratosphere, and the respective regressed O3 anomalies'. Please, could you indicate when a figure is about observed or model-calculated variables? Also, is it monthly means for all the analysis (as indicated in one figure)? Give more information about the time resolution and data processing/filtering of the different analyses.

**We have added more details in Section 3.1 to achieve greater clarity. (Page 5, line 24-30)**

l11, p5: Can be quantitative about the amount of variability captured by the regression? just provide the values of the determination coefficient R2.

**We now present $R^2$ in the new Table 2. We have also added some related discussions in Section 3.1. (Page 5 line 30 to page 6 line 3)**

l19, p5: "project onto" ? replace by "are mostly driven by"

**Changed.**

l26, p5: se comment above about Tsurf missing from the regression.

**Surface temperature (Tsurf) has now been added in the regression model (Eqn. (1)), and the text has been modified accordingly (Section 3.1).**

l29, p6: it's very unlikely that temperature is the driver of O3 variability in the lower stratosphere. The slowing down of O3 destruction by stratospheric cooling (via Chapman cycle) occurs in the upper stratosphere. The ozone budget and variability in the mid-latitude lower stratosphere is dominating by dynamics. I think temperature changes simply reflect dynamical changes that drive O3 variability.

**Agree. We have modified the text accordingly. Now it reads "In the stratosphere above 15 km, negative O$_3$ trends become weaker and are not significant at the 95% confidence level. Long-term changes in O$_3$ in this region are influenced by changes in dynamics (which are reflected in changes in stratospheric temperatures linked to O$_3$ changes), and O$_3$ depletion/recovery that is governed by changes in O$_3$ depleting substances; such changes seem to have cancelled out the decrease in O$_3$ resulting from the tropopause height increase in this region." (Page 7 lines 31-34 to page 8 line 1)**

l9, p8: Difficult to expect a CCM to reproduce specific short-term O3 anomalies. Those anomalies are often driven by specific dynamical variations. Comparisons between CCM simulations and observational time series make more sense when considering long term trends. I would suggest to add the analysis of a REF-C1SD simulation (wind/temperature forced by meteorological analyses instead of being calculated by the model) and compare to REF-C1. It would give an estimate about the effect of model-calculated dynamics/meteorology and biases (yes, there are some) on O3 variability, including trend, for the considered site. Even on long-time series, this could be significant. The authors would be in a stronger position in their attribution analysis. It is less of a problem when considering large-scale averages or multiple sites but here the analysis is limited to a specific site.

**We agree with the reviewer that a free-running CCM has limitations w.r.t. reproducing short-term O3 anomalies. Unfortunately, we have not performed specified dynamics (SD) simulations for comparison with our RED-C1 simulation presented here. However, we did assess simulations by other CCMI models (REF-C1 and REF-C1SD) w.r.t. how their ozone compares to the Lauder ozone record. These other REF-C1 simulations often compare worse to the record than our own simulation. The REF-C1SD simulations usually better reproduce the O$_3$ variability in the stratosphere than REF-C1, but with mixed performance in the troposphere.**

**Therefore, for brevity and clarity, we have decided not to include SD simulations from different models here. The focus of this paper is on attribution of anthropogenic forcings to O$_3$ trends at Lauder, and we believe that this is best achieved by contrasting sensitivity simulations to the reference simulation in a consistent model setup. Moreover, SD simulations are not suitable for use in the attribution study, due to externally imposed meteorological forcing, and in some cases are not self-consistent (r/f Hardiman et al. 2017). Free-running simulations have the merit being more internally self-consistent, taking into account both dynamical and chemical feedbacks, and are more suitable for assessing long-term impact. We have now noted in the text that free-running CCMs have limitations in re-producing O$_3$ variability and trends at one specific site. Furthermore, a comparison between free-running and SD simulations would be more meaningful in a multi-model and a multi-site context that will avoid any potentially biased conclusion, which is outside the scope of this study. We have added some discussions in the text. We hope this is satisfactory. (Page 9 lines 22-29)**

**We very much concur with the reviewer's point that it is important to compare free-running and SD simulations in reproducing O$_3$ trends and variabilities in general, and we will pursue a following-up study to more comprehensively analyse multi-model results for more locations.**

Figures : Can the authors add in all the figure captions which curves are what?

**We have now added the description of each curve in respective figure captions.**

---

## Author Comment (AC2) · 28 Jul 2017

Review of Zeng et al. Attribution of recent ozone changes in the Southern Hemisphere mid-latitudes using statistical analysis and chemistry-climate model simulations

This study uses statistical analysis to understand recent changes in ozone up to 25 km at Lauder, NZ and relate them to changes over similar and longer time periods in chemistry-climate model simulations. Overall the manuscript is clearly written and analysis explained well and I think it would be of interest to the ACP community and would recommend its publication after the authors take some mostly minor comments into consideration.

**Thanks very much for the reviewer's positive comments.**

General comments:

I have concerns about using a chemical measure (O3) for tropopause height to evaluate the O3 field itself. Can you demonstrate that this measure is the same as a PV or temperature based tropopause height over varying timescales?

**We agree that PV would be the ideal quantity to define the tropopause. Unfortunately, we don't have observed variables to calculate PV. We have modified our tropopause height definition to that of Bethan et al. (1996), based on the vertical ozone gradient, which is consistent with the PV definition (e.g., Beekmann et al. 1994). We have recalculated the tropopause height and have incorporated the new tropopause height in our regression analysis. Consequently, the trend in tropopause height at Lauder has changed from 14.2 m/yr to 17.9 m/yr. We have also calculated the tropopause height based on the WMO lapse rate definition, and the result is similar to that based on the ozone definition (trend: 16.5 m/yr). There is no significant systematic difference between the calculated ozone tropopause height and the thermal tropopause height at Lauder. We have revised the relevant text (page 3, lines 24-28), figures (2, 3, and 4) and tables (2 and 3) based on the new tropopause definition.**

I can understand on shorter timescales how the level of o3 of 150 ppb would reflect dynamical variability in tropopause height (although one could argue that a different concentration _100 ppb might be a better choice) but on longer timescales it seems like the chemical changes (through ODS changes) would also be reflected in this quantity. Can you quantify and separate and include some discussion in the text of this issue?

**We have now adopted the new definition of tropopause height based on ozone gradient greater than 60 ppbv/km, constrained to ozone mixing ratios greater than 80 ppbv and exceeding 110 ppbv immediately above the tropopause (Bethan et al., 1996). Longer time-scale $O_3$ changes will inevitably affect the tropopause movement through their feedbacks to the dynamics, and it is not straightforward to separate the influence from chemical changes and those from dynamical changes. We are confident that the new ozone tropopause definition adopted here is a robust choice, and is also consistent with the thermal tropopause height in this case. Moreover, $O_3$ changes in the tropopause region over this period (1987-2014) at Lauder are around 17-25 ppbv, which is significantly smaller than the vertical $O_3$ gradient in this region (typically 50-70 ppbv/km).**

What does chemical O3 tropopause changes look like over the full ref-C2 run can you see the impact of ODS changes causing o3 loss and recovery reflected in this quantity? If you can't see this impact it would be a good demonstration that it is essentially a dynamics only representation over both short and long time-scales. If you do it can be quantified and related to the smaller change in ODSs over the Lauder record

**Using the revised ozone tropopause definition, the calculated trend in tropopause height from the REF-C2 simulation averages about 1.2 m/yr over 1960-2090, and does show some difference between the ozone depletion (4.1 m/yr in 1960-2000) and recovery periods (0.9 m/yr in 2000-2090). Ozone changes do affect dynamics through feedbacks, so we see a larger than average change in tropopause height during the ozone depletion period. This is just for clarification; we do not intend to discuss this in great detail in the present paper.**

I understand that the Lauder record 1987-2014 includes only a modest change in ODSs peaking in the 1990s with little net change so a linear trend might be expected to be flat but it doesn't rule out a some ozone loss in the early period and gain in the later period.

**Both REF-C1 and REF-C2 show some loss of stratospheric $O_3$ from 1990-2000 and recovery from 2000 onwards (Figures 9 and 10). So there is some $O_3$ loss in the early period of the record and gain in the later period. But it is hard to ascertain such a signal from the $O_3$ sonde record at Lauder, due to large variability in observed stratospheric $O_3$. Notes have been added in the text (Page 12, lines 5-10)**

Can you explain why the variables in Table 3 are in some cases different than those shown in Figure 3 like surface Temperature which seems to be highly correlated to lower troposphere ozone is not shown in figure 3.

**We have added surface temperature as a regression function and related discussions, and have modified Tables 2 and 3, and Figures 2 and 3. (Sections 2.2 and 3.1)**

p5 line 17-18 it seems a bit circular argument if you are using o3 to define the tropopause height.

**We have changed the definition of tropopause to be based on the vertical ozone gradient, and the statement stands.**

p10 line1-3 Can you make the same claim for ODS changes over this time refC1 time period (difference between RefC1 and fODS).

**We did not perform a "SEN-C1-fODS" scenario which could be compared to the REF-C1 simulation, therefore cannot directly compare between them. The SEN-C2-fODS simulation cannot be directly compared to the REF-C1 simulation due to differences in model constellation between them.**

p10 line 34 p11 line 1-2 it is not obvious why this should be the case, can you do any additional analysis to explain possible mechanisms.

**Unfortunately, we cannot explain the response of stratospheric $O_3$ trend changes to changes in $O_3$ precursors, which are mainly originated from the troposphere, based on current diagnostics and model simulations. We suspect it is caused by dynamical changes through chemical feedbacks to radiation. We plan to follow up with a more objective analysis. We have added a note to this effect in the text. (Section 4.4; Page 12 lines 14-16)**